# Effect of Radiotherapy in Addition to Surgery in Early Stage Endometrial Cancer: A Population-Based Study

**DOI:** 10.3390/cancers12123814

**Published:** 2020-12-17

**Authors:** Daniel Medenwald, Susan Langer, Cornelia Gottschick, Dirk Vordermark

**Affiliations:** Department of Radiation Oncology, Martin Luther University Halle-Wittenberg, 06120 Halle, Germany; susan.langer@uk-halle.de (S.L.); cornelia.gottschick@uk-halle.de (C.G.); dirk.vordermark@uk-halle.de (D.V.)

**Keywords:** endometrial carcinoma, population-based analysis, radiotherapy, endometrial cancer

## Abstract

**Simple Summary:**

Endometrial cancer is a relatively common tumor among women above the age of 50. The treatment of endometrial cancer in an early stage (FIGO I) is still under dispute. The standard treatment of surgery also includes radiotherapy in certain risk groups. In our study, we assessed the value of radiotherapy (external beam or brachytherapy) on survival in a large population-based cohort of early endometrial cancer cases (FIGO I), with operative treatment respecting known risk factors. The data are based on the epidemiological cancer registries (Robert Koch Institute) from 2000–2017 and encompassed 12,718 cases. We found that radiotherapy is especially beneficial when the tumor infiltrates the muscular tissue (myometrium) of the uterus to a large extend. Radiotherapy was also advantageous in all histopathological grades (cell differentiation). Future studies need to compare different treatment options in an experimental prospective design.

**Abstract:**

Background: The role of radiotherapy in the management of early (FIGO I) endometrial cancer is controversial with limited availability of prospective data from randomized trials. Methods: German Epidemiologic Cancer Registries provided by the Robert Koch Institute. We considered FIGO I cases with recorded operative treatment (*n* = 12,718, 2000–2017). We computed hazard ratios (HR) from relative survival models in relation to the mortality of the general population with 95% confidence intervals (CI). Multivariate models were adjusted for age, stage (IA vs. IB), grading, and chemotherapy. Radiotherapy included external radiotherapy and brachytherapy. Results: Cases with a favorable risk profile (FIGO IA, G1/G2) had a slightly lower survival rate, relative to the general population (FIGO IA: 0.9, G1: 0.91). The proportion of FIGO IA cases was lower in the radiotherapy group (52.6%) vs. cases without radiotherapy (78.6%). Additional treatment with radiotherapy was beneficial in FIGO IB (HR = 0.74) and all histopathological grades, but not FIGO IA cases (HR = 0.93) cases. Compared to IA tumors, IB cases had a HR of 1.51 (95% CI: 1.34–1.7). Conclusions: Radiotherapy in addition to surgery is beneficial for patients in a FIGO IB stage. Further studies need to address the impact of new techniques and risk assessment.

## 1. Introduction

In industrialized countries, endometrial cancer (EC) is the fourth most frequent cancer entity in women [1,2]. The global incidence of EC varies from 4.1/100,000 in less developed countries to an average of 34.1/100,000 in North America [3]. The incidence in Germany is 11,090 incident cases with a relative 5-year survival rate of 78% [4]. 

The risk of getting endometrial carcinoma increases with age (mean age at diagnosis: 69 years) [4]. Further risk factors for the occurrence of EC are hormone therapy with estrogens alone (without progestin protection) [5], late menopausal age [6], diabetes mellitus [7], a high BMI [8], and a positive family history [9]. In contrast, oral contraceptives [6], higher maternal age at birth, late menarche age, and physical activity [10,11] reduce the risk of EC. Approximately 50% of women worldwide and about 70% in Germany with EC show an early stage at diagnosis [4].

NCCN [12] and German [13] guidelines recommend individualized surgery and adjuvant treatment, depending on the stage, histologic subtype, degree, and lymphovascular space invasion [14]. Adjuvant therapy is already recommended at FIGO I (S3-Leitlinie 2018) and consists mainly of vaginal cuff brachytherapy (VBT) in cases with stage IB and/or a grade 3. The adjuvant treatment might also include external beam radiotherapy in cases with lymph-vascular space, or lymph node invasion, or in selected high-risk cases with unknown lymph node status [13,15]. Approximately 50% of women with early-stage endometrial cancer are treated with surgery alone [16].

### 1.1. Scientific Status 

The PORTEC-1 trial has specifically focused on high- or intermediate-risk stage I EC comparing pelvic radiotherapy with no adjuvant treatment EC [17]. Survival was comparable between both treatment arms, while radiotherapy caused a significant reduction in locoregional relapse. In the PORTEC-2 trial, clinical outcomes (vaginal recurrence, locoregional relapse, survival) were not significantly different between patients receiving vaginal brachytherapy or pelvic external beam radiotherapy [18]. Similarly, the GOG-99 trial compared pelvic radiotherapy with no additional treatment to surgery and again found no evidence for an effect of radiotherapy on all-cause mortality [19]. Two other studies balancing VBT between both treatment modalities were included in a meta-analysis by Kong et al. [20]. Both studies (the ASTEC/EN.5 trial [21] and Sorbe et al. [22]) again found no significant survival effect of radiotherapy. Studies based on cancer registries found a high rate of relapses in grade 3 (G3) tumors after adjuvant radiotherapy [23]. To our knowledge, there are no population-based studies that address the effect of adjuvant radiotherapy (external beam and/or brachytherapy) on the survival of early-stage EC. Moreover, this is the first population-based study addressing EC in Germany. 

### 1.2. Objective

We aim to assess the effect of radiotherapy as an additional treatment to surgery in FIGO I cases, in relation to known risk factors of local invasion and grading.

## 2. Results

We found 12,718 cases to be eligible for further analyses (Appendix A). The descriptive findings revealed that the majority of cases were at an earlier stage (IA vs. IB) at the time of diagnosis in the group with and without radiotherapy (78.6%, vs 52.6%, Table 1). However, the proportion of FIGO IA cases was higher in the group without radiotherapy. Likewise, the cases who received radiotherapy had a higher histopathological grade than cases without radiotherapy. Only a small minority of cases received chemotherapy. Coming to survival analyses, we found a good survival rate with a five-year overall survival rate close to 90% in both groups. 

Relative survival was worse than in the reference population, with effect estimates below one in all considered subgroups (Figure 1). The term relative survival relates the mortality of cancer patients to the mortality of the general population, and thus describes the cancer-related mortality rate. In terms of interpretation, it is similar to the outcome of cancer-specific survival. The relative survival was especially low in the subgroup with grade 3 tumors (10-year relative survival: 0.61, 95% CI: 0.59–0.64) and a stage IB tumor (10-year relative survival: 0.68, 95% CI: 0.66–0.69). In univariate Kaplan-Meier analyses, cases with radiotherapy experienced better survival than cases with surgery only (HR = 0.92, 95% CI: 0.84–0.99, Figure 2). 

In the regression models, we found that radiotherapy was associated with survival in Model 5 (HR = 0.84, 95% CI: 0.77–0.92, Table 2) when we adjusted for tumor grade. As in the previous analyses, a stage IB and a grade 3 tumor were associated with a more disadvantageous survival prospect. When we added an interaction term of T-stage and the application of radiotherapy to the survival model, we found the advantageous effect of radiotherapy to be limited to cases with a stage IB (HR of interaction: 0.8 95% CI: 0.67–0.95, Model 6). This notion is supported by the stage-specific Kaplan-Meier plot (Figure 3). There was no similar interaction of T-stage with radiotherapy in the regression model. Thus, radiotherapy was beneficial in all three histopathological grades, which was, however, only statistically significant in grade 2 and 3 malignancies (Figure 4).

When we examined risk groups, we observed the highest mortality risk in the FIGO IB, G3, subgroup while radiotherapy reduced mortality mostly in respective cases with G1/2 grading (HR = 0.72, 95% CI: 0.58–0.91, Table 3) with a slightly lower effect in G3 cases not reaching statistical significance (HR = 0.85, 95% CI: 0.61–1.21, Table 3). 

### Sensitivity and Additional Analyses

In the respective additional analyses, we found virtually no change in the associations found in previous models when we used an additive or multiplicative relative survival model, respectively. Likewise, there was no numerical change in the association of radiotherapy with survival when we incorporated all FIGO I cases (Appendix A). Missing information on treatment was not associated with either age or year of diagnosis when we presumed a level of significance of 5%. In the sensitivity analysis restricted to endometrioid carcinomas, we again found few changes in the computed effect estimates (Appendix A). Effect estimates were similar in cases with and without histopathological examination of lymph nodes (Appendix A).

## 3. Discussion

In summary, we found that radiotherapy is beneficial as an additive to the surgical treatment of early-stage endometrial cancer. However, in our data, this relation was considerably stronger in cases with a stage IB, respectively. In contrast, the group with radiotherapy showed a better survival rate than the group without radiotherapy in all histopathological grades. Compared to the general population, the survival in cases with a favorable risk profile (T1a and grade 1) was only slightly lower than in the general population. 

As the development of endometrial cancer is associated with obesity [24], which itself is a strong risk factor for an impaired survival prospect [25,26], the slightly reduced mortality in cases without major myometrial involvement and G1/G2 differentiation compared to the general population is unsurprising. 

Regarding the effect of radiotherapy, we found that the group with radiotherapy had a better survival prospect in relation to the general population than the group without this treatment option. The aforementioned Cochrane meta-analysis concluded that there is no evidence for a survival benefit of radiotherapy after surgery in early-stage tumors. However, as the authors stated, this analysis was severely underpowered with only the GOG-99 [19] and PORTEC-I [27] trial entering the final analysis. Both studies revealed an advantage of adjuvant radiotherapy when local recurrence is concerned, but not in terms of overall survival. However, in both studies, investigators failed to power the sample size sufficiently in order to detect a possible difference in survival. The PORTEC-I trial found that a myometrial invasion >50% is a risk factor for a disadvantageous survival, although not reaching a statistically significant level. The GOG-99 study observed a lower survival in the group without adjuvant treatment in the predefined high-risk group, which was similar to the low-risk group when the patients received radiotherapy. This is similar to our findings where a higher mortality risk in cases with a deep myometrial invasion experienced a better survival when they received radiotherapy as an additional treatment. For the second important risk factor of tumor grade, radiotherapy was beneficial in all grades. Again, in the PORTEC-I trial, a grade 3 tumor showed an almost five-fold risk increase compared to grade 1. Unfortunately, the authors failed to perform analyses stratified in relation to stage or grade, which makes it difficult to assess the effect of radiotherapy in the context of both risk factors.

Furthermore, these findings also need to be interpreted in the light of an experimental study design, which might not concur with a real-life environment. Patients from a lower socioeconomic background might not enter a clinical trial in the first place [28], while this group is especially prone to missed follow-up visits, and thus relapses [29]. In such circumstances when a proper re-staging and follow-up is difficult to realize because of poor compliance [30], a treatment using radiotherapy might be especially advantageous. Such relations might explain the difference between our findings and the results of prospective trials. 

The early PORTEC-I trial, which started in 1990 and ended in 1997, used 2D planning techniques. With the advent of 3D techniques, we are now able to limit the radiation dose to organs at risk, thus avoiding side effects. As our data encompass the period since 2000 when 3D planning became the standard approach. less advanced techniques are unlikely to obscure a potential benefit by the application of radiotherapy. 

Similarly, an older Norwegian study found that external pelvic radiotherapy after brachytherapy had no association with overall survival, but resulted in improved local control. On the other hand, distant metastases were more frequent in the group with external radiotherapy when compared to the control group with mere observation [31]. 

In our analysis, we found that radiotherapy was advantageous in the risk group of FIGO IB G3 cases, and the intermediate group of FIGO IB G1 or G2 cases. Based on registry data, a later study reported high rates of locoregional relapse, even after radiotherapy [23]. Such an adverse risk related to grade 3 tumors was also present in our data. However, as the interaction analysis revealed, radiotherapy fails to compensate for this adversity. 

Data from the SEER database reported a considerable increase in the utilization of vaginal cuff brachytherapy as an adjuvant treatment of early-stage endometrial cancer [32]. However, as our data indicate, the application of radiotherapy in cases with superficial myometrial involvement is of little benefit. Thus, a careful selection of cases profiting from adjuvant treatment is a major challenge. Other data from population-based cohorts report similar findings. AlHilli et al. [33,34] found that VBT improved survival in stage IA G3 and IB G1/G2 and G3, in data from the National Cancer Database. However, external beam radiotherapy was inferior to VBT, apart from cases with a stage of IB G3. Another study, based on the same database, similarly reported improved survival with the application of VBT in stage IA and IB [35,36].

In our data, we adjusted for chemotherapy, where we observed higher mortality in cases with chemotherapy, compared to cases without this treatment. Such a treatment is uncommon for early-stage endometrial cancer and might be restricted to cases with a rapidly progressing disease. However, only a few cases received chemotherapy, amounting to no more than 3%. With the available data, we cannot infer the indication for chemotherapy, which restricts further analyses.

Two prospective randomized trials analyzed sequential adjuvant chemotherapy and radiotherapy and found a 36% reduction in the combined endpoint of death or relapse, while overall survival was not affected [37].

Finally, as we performed a population-based analysis, our findings apply directly to an unselected collective where the treatment effect becomes directly apparent. As the number of cases with endometrial cancer rose in recent years, the population effect of treatment interventions is especially relevant from a public health perspective [38]. Here, the rise of non-endometroid cancers might pose a challenge to future treatment guidelines [34,38]. 

### Limitations

The most important limitation of our study is a lack of information on the form of radiotherapy applied, which might be vaginal brachytherapy (VBT), or external beam radiotherapy (EBRT). However, when it comes to overall survival the PORTEC-II trial found a similar survival in both treatments [27]. Thus, effect estimates might still hold for radiotherapy as a whole while we cannot make statements on the possible benefits of VBT or EBRT. 

Another source of bias might derive from missing data, which we recorded in 7% of all considered cases when ‘treatment’ was concerned. However, when we analyzed the association with either age or year of diagnosis, we found no significant associations, making a serious bias unlikely. In our analysis, we only considered cases from registries with higher data quality. Despite this, some cases might not have entered the registry at the time of diagnosis. However, as we conditioned our data set to cases with a FIGO I stage and respective histopathological information by predefined inclusion criteria, we would only expect biased effects if the tumor biology of cases not enrolled differs from cases used for further analyses. Furthermore, as we adjusted for age, any bias related to missing data caused by an age effect is thus accounted for, if the assumption of missingness at random holds. 

In recent years, new molecular parameters have increasing importance in the risk stratification of endometrial cancer [36]. The design of the Portec-4a study respects the molecular-integrated risk profile through a comprehensive stratification process [27]. These factors might employ a more refined risk assessment, but cannot be analyzed in our study. To give an example, the POLE mutation represents a favorable survival prospect [27]. In this trial, the standard treatment of VBT is reserved for an intermediate risk profile, while the low-risk group receives no adjuvant radiotherapy. Another important risk factor is a mutation of TP53, in relation to poor prognosis. In this scenario, platinum derivatives might constitute a vital targeted therapy [36]. Results relating the experimental risk assessment to the standard treatment might change the paradigms profoundly.

## 4. Materials and Methods 

### 4.1. Data and Material

We used data from the German Epidemiological Cancer Registries, provided by the Robert Koch Institute (RKI) for public use. This population-based registry incorporates federal data that respective state transfer to the RKI. The dataset contains, amongst other information, data on grading and histology, TNM-stage, cause and date of death, date of birth and date of diagnosis (month as the smallest temporal unit in each date variable), and treatment. However, information on performance status, comorbidities, or further details on treatment procedures, such as the administered radiation dose or fractionation is not included. In addition, the TNM-stage in this dataset refers to the clinical or pathological stage (if an operation was performed), while there is no information on the clinical stage in cases where a p-stage is available [39].

The proportion of cases diagnosed from death certificate only was below 13% (with the largest proportion found in Hesse of 8.3%) in all considered states as recommended by Rossi et al [40]. To ensure adequate data quality, we only included cases from states with a proportion of missing information on treatment below 50% which were Bavaria, Thuringia, Mecklenburg-Western Pomerania, Saxony-Anhalt, Brandenburg, Saxony, and Berlin (see Figure 5 for the inclusion process). 

In our analyses, we considered cases diagnosed between 2000 and 2017, and documented surgical treatment. During this period, the TNM coding system changed encompassing TNM version 5 to 8. For TNM version 5 and 6, we considered the stages T1a and T1b as equivalent to the T1a of the TNM 7 and 8. Likewise, we regarded T1c (version 5 and 6) as corresponding to T1b (version 7 and 8). We only considered FIGO I cases (T1a-T1c, N0/x, and M0/x) with an age at diagnosis of 18 or older for further analyses. We excluded cases diagnosed from death certificates only (DCO) or with a diagnosis from an autopsy. 

In our study, the term radiotherapy refers to any form of radiotherapy, without differentiation between brachytherapy and external beam radiotherapy.

Cases with the histological grade coded in accordance with the ADT (Arbeitsgemeinschaft Deutscher Tumorzentren) were also excluded. We considered an ICD-O3 of 8441, 8950, 8980, and 8310 as a grade 3 tumor. 

### 4.2. Statistical Analyses

We used relative survival regression models (rstrans function in R of the relsurv package) with German mortality tables (accessible via https://www.mortality.org/) as the reference population, using a transformation approach to compute effect estimates [41]. In essence, this approach computes the expected survival if the case was representative of the reference population [42]. In the results section, we refer to the estimates as ‘hazard ratios’ (HR).

Apart from this approach, an Andersen multiplicative model can serve as the estimand for relative survival regression models, resulting in estimates on a multiplicative scale. This method is an extension of the Cox model. As a third alternative, additive models can compute effect estimates on an additive scale. We considered both approaches (multiplicative and additive) as a sensitivity analysis, applying the Estève method for additive models [41]. Multivariate models of the association of treatment by radiotherapy with relative survival were adjusted for grading, T-stage (T1a/b according to TNM version 7/8), age at the time of diagnosis, and chemotherapy. In an analysis of risk strata, we defined the following subgroups according to FIGO stage and grading (in brackets): IA (G1/G2), IA (G3), IB (G1/G2), and IB (G3). 

For relative survival, we applied the Pohar Perme approach [43] as it is implemented in the rs.surv function of the same package in R. In the statistical analyses, we used different models with a successively increasing number of covariates to estimate the explanatory power of each parameter on the effect of the association between radiotherapy and relative survival.

We obtained German mortality tables from the Human Mortality Database (available at https://www.mortality.org/). We computed the respective effect estimates with 95% confidence intervals. All analyses were performed in R version 3.6.0 [44].

### 4.3. Additional Analyses

In addition, we performed an analysis of missing treatment data, considering the whole data set. In this analysis, we incorporated ‘missing treatment’ as the outcome variable in a logistic regression model with age and year of diagnosis as predictors. We performed another sensitivity analysis where we included all FIGO I cases irrespectively if they allowed for a differentiation between FIGO IA and IB (T1 as an additional category in multivariate regression models). We performed another sensitivity analysis in which we only considered endometrioid carcinomas (ICD-O3 morphology: 8-380). Finally, we performed another analysis where we differentiated between cases with (pN, lymphadenectomy) and without (cN) histopathological examination of lymph nodes.

## 5. Conclusions

In conclusion, we found that radiotherapy in addition to surgery benefits mostly patients in a FIGO IB stage, irrespective of the histopathological grade in a population-based sample. Further studies need to address the impact of new radiotherapy technologies, such as intensity-modulated radiotherapy before the background of a changing risk pattern.

## Figures and Tables

**Figure 1 cancers-12-03814-f001:**
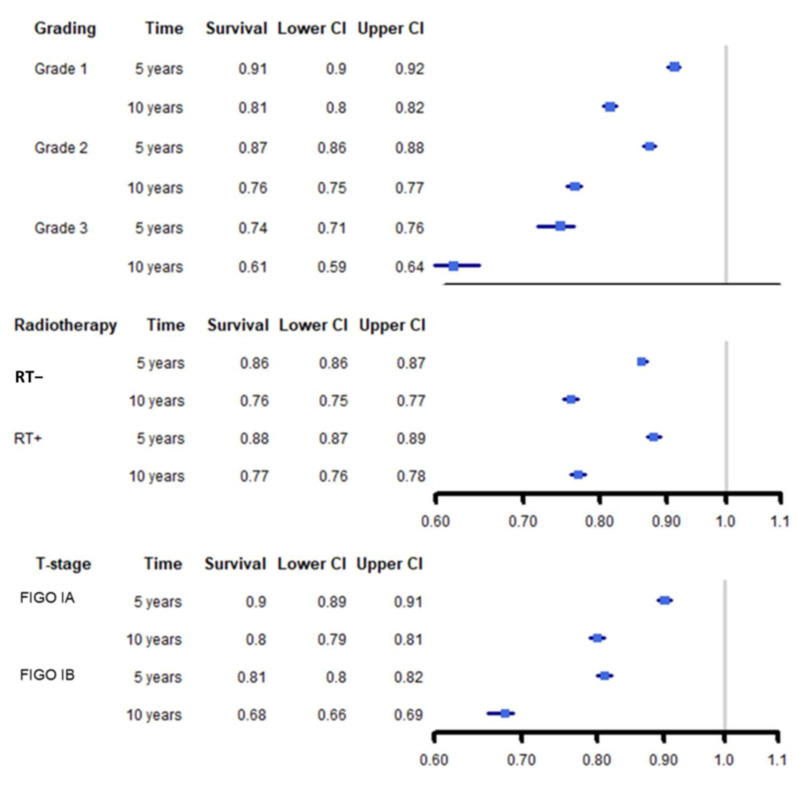
Forest plot on relative survival in relation to the mortality of the general population. Black lines: 95% confidence intervals, blue boxes: Effect estimates.

**Figure 2 cancers-12-03814-f002:**
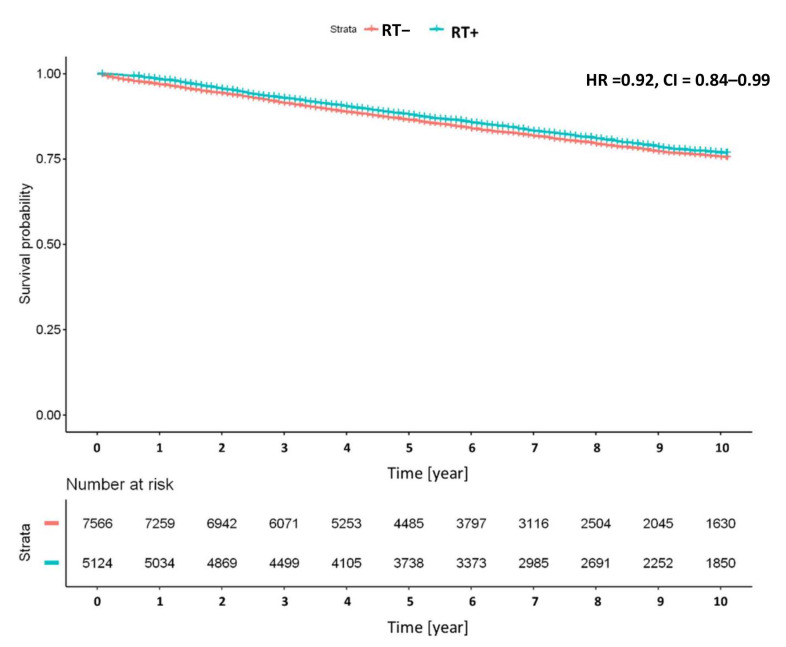
Kaplan-Meier survival plot of cases with and without radiotherapy, HR = hazard ratio with 95% confidence intervals.

**Figure 3 cancers-12-03814-f003:**
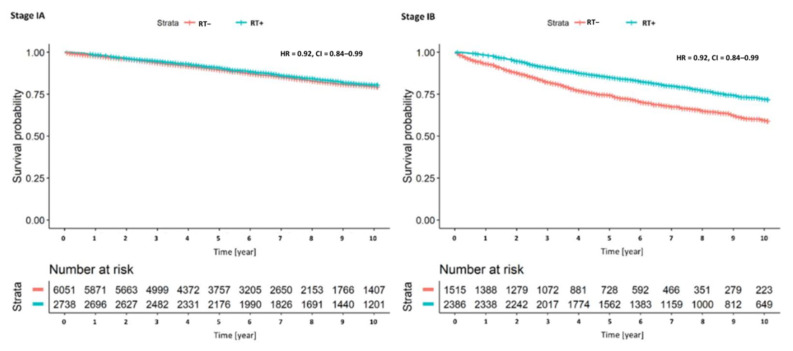
Stage-specific Kaplan-Meier survival plot (Stage IA and IB) of cases with and without radiotherapy, HR = hazard ratio with 95% confidence intervals.

**Figure 4 cancers-12-03814-f004:**
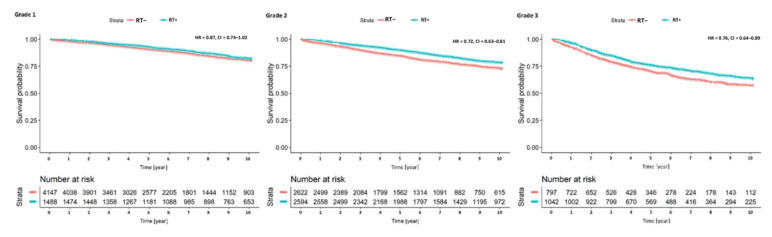
Grade-specific Kaplan-Meier survival plot of cases with and without radiotherapy, HR = hazard ratio with 95% confidence intervals.

**Figure 5 cancers-12-03814-f005:**
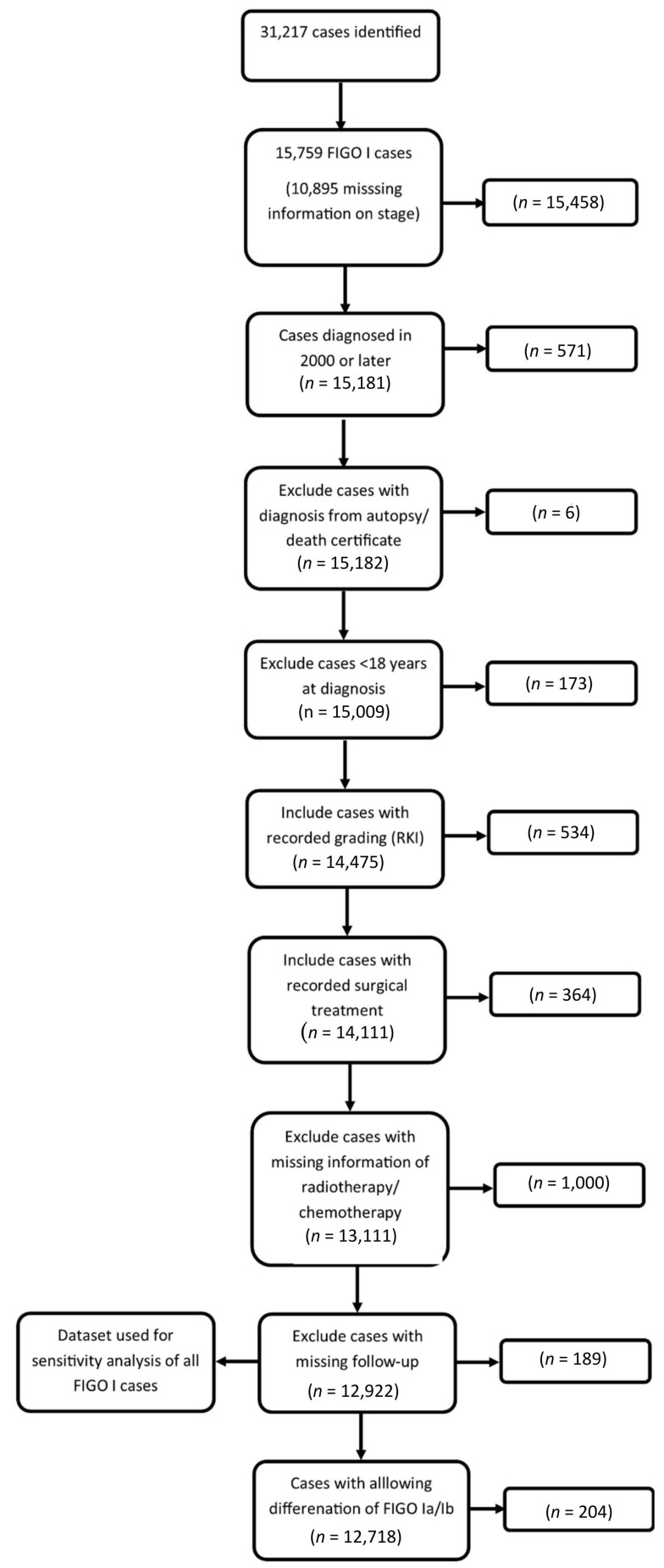
Flow chart of the inclusion/exclusion process.

**Table 1 cancers-12-03814-t001:** Basic characteristics.

Parameter		No Radiotherapy		Radiotherapy	
Stage	IA	6065	78.6	2739	52.6
	IB	1527	19.8	2387	45.9
	T1	122	1.6	79	1.5
Grade	1	4211	54.6	1514	29.1
	2	2683	34.8	2631	50.5
	3	822	10.7	1061	20.4
Chemotherapy	No	7599	98.5	5086	97.7
	Yes	117	1.5	120	2.3
Age (mean, (95% CI))		53.6 (53.3–53.9)		53.3 (53.0–53.6)	
Survival (%, (95% CI))	5 year	86.2 (85.4–87.1)		88.0 (87.1–88.9)	
	10 year	75.5 (74.2–76.7)		76.7 (75.7–78.1)	

**Table 2 cancers-12-03814-t002:** Results of the relative survival model.

Parameter	Model 1	Model 2	Model 3	Model 4	Model 5	Model 6
RT–	Ref.	Ref.	Ref.	Ref.	Ref.	Ref.
RT+	0.93 (0.85–1.01)	1.01 (0.93–1.1)	0.92 (0.84–1)	0.84 (0.77–0.92)	0.84 (0.77–0.92)	0.93 (0.83–1.05)
Age (10 years)		1.08 (1.07–1.08)	1.08 (1.07–1.08)	1.07 (1.07–1.08)	1.07 (1.07–1.08)	1.07 (1.07–1.08)
Stage IA *			Ref.	Ref.	Ref.	Ref.
Stage IB **			1.42 (1.3–1.55)	1.36 (1.25–1.48)	1.36 (1.25–1.48)	1.51 (1.34–1.7)
Grade 1				Ref.	Ref.	Ref.
Grade 2				1.24 (1.12–1.37)	1.24 (1.12–1.36)	1.22 (1.11–1.35)
Grade 3				2.26 (2.02–2.53)	2.2 (1.96–2.46)	2.17 (1.94–2.43)
CT–					Ref.	Ref.
CT+					1.81 (1.38–2.36)	1.81 (1.39–2.37)
Stage/RT Interaction ^†^						0.8 (0.67–0.95)

Model 1: crude model; Model 2: age; Model 3: age, stage (IA vs. IB); Model 4: age, stage (IA vs. IB), grading; Model 5: age, stage (IA vs. IB), grading, chemotherapy (CT); * Stage IA (TNM version 8); ** Stage IB (TNM version 8), ^†^ T1a and no radiotherapy as the reference for the interaction. Interaction estimate refers to the effect of RT in Stage IB.

**Table 3 cancers-12-03814-t003:** Multivariate analysis of relative survival model in relation to FIGO/grade risk constellation considering all early-stage T1, N0/x, M0/x cases.

Parameter	Model
RT–	Ref.
RT+	0.94 (0.81–1.1)
Age (10 years)	1.08 (1.07–1.08)
IA, G1/2	Ref.
IA, G3	1.64 (1.27–2.12)
IB, G1/2	1.61 (1.4–1.86)
IB, G3	2.71 (2.09–3.52)
CT–	Ref.
CT+	2.29 (1.64–3.19)
RT Interaction ^†^	
IA, G1/2	Ref.
IA, G3	1.23 (0.85–1.76)
IB, G1/2	0.72 (0.58–0.91)
IB, G3	0.85 (0.61–1.21)

^†^ Interaction estimates refer to the effect of RT in Stage IA, G1/G2.

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
