# Peer review of "Effect of Radiotherapy in Addition to Surgery in Early Stage Endometrial Cancer: A Population-Based Study"

_cancers, 2020, doi:10.3390/cancers12123814_

Round 1

Reviewer 1 Report

Introduction section:

Page 2, Line 54: PORTEC-1 ....focused "on High-Intermediate-risk stage I patients".

Include a commnent of PORTEC-2 trial.

Material and methods section:

Do the authors know the indications for chemotherapy in these group of patients.

Discusion section:

Page 7 line157:  Portec-1 included patients in stage IA G2-3 and IB G1-G2, so considering what we know at the present there was no sense in stratifiying patients by stage or grade.

Author Response

Page 2, Line 54: PORTEC-1 ....focused "on High-Intermediate-risk stage I patients".

Include a commnent of PORTEC-2 trial.

A respective comment was added to the manuscript.

Material and methods section:

Do the authors know the indications for chemotherapy in these group of patients.

Unfortunately, this cannot be deduced from the available data. We mentioned this as a limitation in the discussion section:

With the available data, we cannot infer the indication for chemotherapy, which restricts further analyses.

Discusion section:

Page 7 line157:  Portec-1 included patients in stage IA G2-3 and IB G1-G2, so considering what we know at the present there was no sense in stratifiying patients by stage or grade.

As our data indicate treatment effects might be different in the respective stages which might result in different conclusion in relation to these parameters. This is the reason why we discussed stratifying cohorts.

Reviewer 2 Report

This is an interesting manuscript as it is a comprehensive report on the benefit of post-operative adjuvant radiotherapy in early stage endometrial cancer. Compare with previous studies such as PORTEC-1 and GOG-99 trials which compare pelvic radiotherapy with no adjuvant treatment in early stage EC. This is the first studies addressing the effect of adjuvant radiotherapy on overall survival of early stage EC. However, there is one limitation in this study is the lack of information on the form of radiotherapy technique applied. Whether if its vaginal brachytherapy or external beam radiotherapy was utillized on each group. Nevertheless, this manuscript will add to a growing knowledge of the beneficial effect of adjuvant radiotherapy in addition to surgery for early stage endometrial ca. Hopefully, this will eventual lay as a foundation for the future treatment in early stage (FIGO1 A-B, grade 1-3) endometrial cancer.

Author Response

This is an interesting manuscript as it is a comprehensive report on the benefit of post-operative adjuvant radiotherapy in early stage endometrial cancer. Compare with previous studies such as PORTEC-1 and GOG-99 trials which compare pelvic radiotherapy with no adjuvant treatment in early stage EC. This is the first studies addressing the effect of adjuvant radiotherapy on overall survival of early stage EC. However, there is one limitation in this study is the lack of information on the form of radiotherapy technique applied. Whether if its vaginal brachytherapy or external beam radiotherapy was utillized on each group. Nevertheless, this manuscript will add to a growing knowledge of the beneficial effect of adjuvant radiotherapy in addition to surgery for early stage endometrial ca. Hopefully, this will eventual lay as a foundation for the future treatment in early stage (FIGO1 A-B, grade 1-3) endometrial cancer.

We thank the reviewer for the comments.

Reviewer 3 Report

The authors have analyzed around 13000 endometrial cancer (EC) cases from German epidemiological databases and compared their analysis with GOG-99 and PORTEC-1 trials. They concluded that radiotherapy is a better treatment option for EC patients.

  1. However, at present from this patient cohort, it can’t be concluded that radiotherapy is beneficial over progestin therapy or any other therapy at stage I.
  2. The manuscript lacks a clear conclusion and the results are not very supportive.
  3. Line 171-172: The authors argue that radiotherapy is advantageous in G3 cases whereas a greater number of G3 cases are found in the radiotherapy group (Line 72-73).
  4. There is no significant difference in terms of 5 years survival between radiotherapy and no radiotherapy groups (88 vs 86.2).
  5. Most of the sentences are very complex and difficult to follow. lines 204-208,  

Author Response

The authors have analyzed around 13000 endometrial cancer (EC) cases from German epidemiological databases and compared their analysis with GOG-99 and PORTEC-1 trials. They concluded that radiotherapy is a better treatment option for EC patients.

  1. However, at present from this patient cohort, it can’t be concluded that radiotherapy is beneficial over progestin therapy or any other therapy at stage I.

The reviewer is right. However, the main objective of this analysis was the effect of radiotherapy as it is recommended by international guidelines. Other treatment options might be beneficial, but are not the objective of this paper.

  1. The manuscript lacks a clear conclusion and the results are not very supportive.

The conclusion of the manuscript was changed to make the findings clearer.

  1. Line 171-172: The authors argue that radiotherapy is advantageous in G3 cases whereas a greater number of G3 cases are found in the radiotherapy group (Line 72-73).

As we performed multivariate analyses and this is a real life cohort we can conclude that radiotherapy is beneficiary in certain subgroups although treatments are not balanced. However, any artificial alteration of the cohort might lead to seriously biased results.

  1. There is no significant difference in terms of 5 years survival between radiotherapy and no radiotherapy groups (88 vs 86.2).

The reviewer is right. The survival effect results from multivariate models. Univariate analyses as in this example might be confounded and need to be adjusted for. This is especially relevant when the tumour grade is concerned. In addition, survival models also address the temporal course of the survival curve.

  1. Most of the sentences are very complex and difficult to follow. lines 204-208,  

We changed the respective sentences in order to facilitate the readability. We wrote:

The most important limitation of our study is a lack of information on the form of radiotherapy applied, which might be vaginal brachytherapy (VBT), or external beam radiotherapy (EBRT). However, when it comes to overall survival the PORTEC-II trail found a similar survival in both treatments [26]. Thus, effect estimates might still hold for radiotherapy as a whole while we cannot make statements on possible benefits VBT or EBRT.

Reviewer 4 Report

This study by Medenwald D et al, assessed the effect of radiotherapy as an additional treatment to surgery in FIGO I cases in relation to known risk factors of local invasion and grading. The manuscript is straightforward, well written, and concise, and has clear results. Definitely deserves to be published and is a valuable contribution to the “cancersjournal. Some minor flaws need to be addressed before publication.

Minor points:

[1] “1.1. Scientific status:”, Page 2/13:

Beyond PORTEC-1, GOG-99 and ASTEC/EN.5, there is also available the Norwegian trial, published in 1980. 540 women with clinical stage 1 endometrial carcinoma, after hysterectomy and postoperative vaginal brachytherapy (60 Gy to the mucosal surface) were randomly assigned to additional EBRT (40 Gy in 2 Gy fractions) or observation. Although additional EBRT reduced vaginal and pelvic relapse rates (2% at 5 years vs 7% in the control group), more distant metastases were found in the RT group (10% vs 5%), and survival was not improved (89% vs 91% at 5 years). The subgroup with grade 3 tumors with deep (>50%) myometrial invasion showed improved local control and survival after EBRT (18% vs 27% cancer-related deaths); however, there were too few patients in this category to reach significance.

Recommended reference: Aalders J, et al. Postoperative external irradiation and prognostic parameters in stage I endometrial carcinoma: clinical and histopathologic study of 540 patients. Obstet Gynecol. 1980;56:419–427.

[2] “3. Discussion”, Page 6/13:

I recommend to incorporate in the discussion part a comment on the role of combined radiation therapy and chemotherapy. The large NSGO 9501/EORTC 55991 trial, randomly compared EBRT with or without four cycles of chemotherapy in 382 patients. Various chemotherapy combinations were used, given before or after RT. This trial was the first to show a significant 7% increase in progression-free survival with the addition of chemotherapy (79% vs 72%, P = 0.03), but no significant difference in overall survival. In the published pooled data analysis with the Italian MaNGO ILIADE-III trial, results were similar, with a statistically significant difference in 5-year PFS favoring the combined arm (78% vs 69%, P = 0.009), but only a trend for improved 5-year OS (82% vs 75%, P = 0.07).

Recommended reference: Hogberg T, et al. Sequential adjuvant chemotherapy and radiotherapy in endometrial cancer—results from two randomised studies. Eur J Cancer. 2010;46:2422–2431.

[3] “3. Discussion”, Page 6/13:

At the end of the discussion section, I recommend to report that the molecular sub-groups of endometrial cancer provide the basis for a more robust classification of endometrial cancer with prognostic significance. The corresponding overexpression of P53 has been found in 40–50% of endometrial adenocarcinomas, with strong and significant benefit of added adjuvant chemotherapy in patients with P53 mutational expression.

Recommended reference: Boussios S, et al. Ovarian carcinosarcoma: Current developments and future perspectives. Crit Rev Oncol Hematol. 2019;134:46-55.

Author Response

This study by Medenwald D et al, assessed the effect of radiotherapy as an additional treatment to surgery in FIGO I cases in relation to known risk factors of local invasion and grading. The manuscript is straightforward, well written, and concise, and has clear results. Definitely deserves to be published and is a valuable contribution to the “cancers” journal. Some minor flaws need to be addressed before publication.

Minor points:

[1] “1.1. Scientific status:”, Page 2/13:

Beyond PORTEC-1, GOG-99 and ASTEC/EN.5, there is also available the Norwegian trial, published in 1980. 540 women with clinical stage 1 endometrial carcinoma, after hysterectomy and postoperative vaginal brachytherapy (60 Gy to the mucosal surface) were randomly assigned to additional EBRT (40 Gy in 2 Gy fractions) or observation. Although additional EBRT reduced vaginal and pelvic relapse rates (2% at 5 years vs 7% in the control group), more distant metastases were found in the RT group (10% vs 5%), and survival was not improved (89% vs 91% at 5 years). The subgroup with grade 3 tumors with deep (>50%) myometrial invasion showed improved local control and survival after EBRT (18% vs 27% cancer-related deaths); however, there were too few patients in this category to reach significance.

Recommended reference: Aalders J, et al. Postoperative external irradiation and prognostic parameters in stage I endometrial carcinoma: clinical and histopathologic study of 540 patients. Obstet Gynecol. 1980;56:419–427.

We thank the reviewer for pointing this out and included the respective reference in the manuscript.

[2] “3. Discussion”, Page 6/13:

I recommend to incorporate in the discussion part a comment on the role of combined radiation therapy and chemotherapy. The large NSGO 9501/EORTC 55991 trial, randomly compared EBRT with or without four cycles of chemotherapy in 382 patients. Various chemotherapy combinations were used, given before or after RT. This trial was the first to show a significant 7% increase in progression-free survival with the addition of chemotherapy (79% vs 72%, P = 0.03), but no significant difference in overall survival. In the published pooled data analysis with the Italian MaNGO ILIADE-III trial, results were similar, with a statistically significant difference in 5-year PFS favoring the combined arm (78% vs 69%, P = 0.009), but only a trend for improved 5-year OS (82% vs 75%, P = 0.07).

Recommended reference: Hogberg T, et al. Sequential adjuvant chemotherapy and radiotherapy in endometrial cancer—results from two randomised studies. Eur J Cancer. 2010;46:2422–2431.

A respective section was added to the manuscript.

[3] “3. Discussion”, Page 6/13:

At the end of the discussion section, I recommend to report that the molecular sub-groups of endometrial cancer provide the basis for a more robust classification of endometrial cancer with prognostic significance. The corresponding overexpression of P53 has been found in 40–50% of endometrial adenocarcinomas, with strong and significant benefit of added adjuvant chemotherapy in patients with P53 mutational expression.

Recommended reference: Boussios S, et al. Ovarian carcinosarcoma: Current developments and future perspectives. Crit Rev Oncol Hematol. 2019;134:46-55.

A respective statement was added to the main text.

Round 2

Reviewer 3 Report

The authors response seems ok. The conclusion is not significantly improved.